# Novel Biodegradable Composite of Calcium Phosphate Cement and the Collagen I Mimetic P-15 for Pedicle Screw Augmentation in Osteoporotic Bone

**DOI:** 10.3390/biomedicines9101392

**Published:** 2021-10-04

**Authors:** Harald Krenzlin, Andrea Foelger, Volker Mailänder, Christopher Blase, Marc Brockmann, Christoph Düber, Florian Ringel, Naureen Keric

**Affiliations:** 1Department of Neurosurgery, University Medical Center Mainz, 55131 Mainz, Germany; Andrea.Foelger@web.de (A.F.); Florian.Ringel@unimedizin-mainz.de (F.R.); Naureen.Keric@unimedizin-mainz.de (N.K.); 2Max Planck Institute for Polymer Research, 55128 Mainz, Germany; Volker.Mailaender@unimedizin-mainz.de; 3Center for Translational Nanomedicine, University Medical Center Mainz, 55131 Maniz, Germany; 4Personalized Biomedical Engineering Lab, Frankfurt University of Applied Sciences, 60318 Frankfurt am Main, Germany; cblase@fb2.fra-uas.de; 5Department of Neuroradiology, University Medical Center Mainz, 55131 Mainz, Germany; Marc.Brockmann@unimedizin-mainz.de; 6Department of Radiology, University Medical Center Mainz, 55131 Mainz, Germany; Christoph.Dueber@unimedizin-mainz.de

**Keywords:** osteoporosis, osteoporotic vertebral fractures, polymethylmethacrylate, calcium phosphate cement, collagen I mimetic P-15

## Abstract

Osteoporotic vertebral fractures often necessitate fusion surgery, with high rates of implant failure. We present a novel bioactive composite of calcium phosphate cement (CPC) and the collagen I mimetic P-15 for pedicle screw augmentation in osteoporotic bone. Methods involved expression analysis of osteogenesis-related genes during osteoblastic differentiation by RT-PCR and immunostaining of osteopontin and Ca^2+^ deposits. Untreated and decalcified sheep vertebrae were utilized for linear pullout testing of pedicle screws. Bone mineral density (BMD) was measured using dual-energy X-ray absorptiometry (DEXA). Expression of ALPI II (*p* < 0.0001), osteopontin (*p* < 0.0001), RUNX2 (*p* < 0.0001), and osteocalcin (*p* < 0.0001) was upregulated after co-culture of MSC with CPC-P-15. BMD was decreased by 28.75% ± 2.6%. Pullout loads in untreated vertebrae were 1405 ± 6 N (*p* < 0.001) without augmentation, 2010 ± 168 N (*p* < 0.0001) after augmentation with CPC-P-15, and 2112 ± 98 N (*p* < 0.0001) with PMMA. In decalcified vertebrae, pullout loads were 828 ± 66 N (*p* < 0.0001) without augmentation, 1324 ± 712 N (*p* = 0.04) with PMMA, and 1252 ± 131 N (*p* < 0.0078) with CPC-P-15. CPC-P-15 induces osteoblastic differentiation of human MES and improves pullout resistance of pedicle screws in osteoporotic and non-osteoporotic bone.

## 1. Introduction

Osteoporosis is a multifactorial disease characterized by a micro-architectural loss of bone mass and strength [1]. Worldwide, more than 200 million people have osteoporosis, with more than 70% of those being over 80 years of age [2]. According to the World Health Organization (WHO), osteoporosis is defined by a bone mineral density (BMD) T-score of more than 2.5 standard deviations below reference ranges, as measured by dual-energy X-ray absorptiometry (DEXA) [3].

In osteoporosis, impaired bone strength results from either the inability to achieve peak bone mass or an imbalance in bone remodeling [4]. These changes lead to increased skeletal fragility and fracture susceptibility [5]. Up to 80% of the peak bone mass is determined by genetic factors. Genome-wide association studies identified several genes that favor osteoporosis, including low-density lipoprotein receptor-related protein 5 (*LRP5*) and receptor activator of NF-κβ (*RANK*) [6]. Hormonal status during childhood and adolescence, environmental factors, exercise, nutrition, and smoking contribute to the attainable peak bone mass [7]. Bone remodeling is pivotal to maintain structural bone integrity and contributes to a systemic balance of calcium and phosphorus [8]. During this process, numerous genetic markers are expressed that characterize distinct developmental steps [8].

Low-impact fragility fractures resulting from a low BMD are the most critical complication of osteoporosis and lead to a significantly decreased quality of life and increased morbidity and mortality [9]. With more than 3.5 million annual cases within the European Union, osteoporotic fractures impose a substantial economic burden on healthcare systems [10]. Treatment costs are estimated to exceed 37 billion euros, more than 70% of which are allotted to osteoporotic fractures [10]. Vertebral fractures are a common cause of back pain in patients with osteoporosis. In later stages, skeletal deformity, joint incongruity, and tension on muscles and tendons might lead to chronic back pain and disability [11]. Vertebroplasty using polymethylmethacrylate (PMMA) proved to be successful in fracture stabilization and pain relief [12]. In cases of instability, posterior spinal fusion with instrumentation might be necessary. Osteoporosis-related complications such as proximal compression fractures, junctional kyphosis, and instrumentation failure are high in osteopenia and osteoporosis patients [13]. Lower bone density is associated with high non-fusion rates and implant failure [14]. The loss of 1 mm cortical thickness results in up to a 50% reduction in implant strength [15]. To increase fusion rates and prevent implant failure, considerable effort has been spent optimizing preoperative osteoporosis medications. However, the effect on fusion rates and clinical outcome remains ambiguous [16]. In a smaller case series, cement augmentation of pedicle screws (e.g., PMMA) lowered non-fusion rates and reduced implant failure [16,17]. The use of PMMA in osteoporosis is not beneficial in the long term. A strong exothermic reaction during polymerization causes substantial damage to the surrounding bone, further compromising the already impaired bone remodeling balance [18]. As PMMA is not resorbable, it remains in situ, leading to a chronic inflammatory response in some patients, further compromising implant strength and integrity [19]. In a large meta-analysis, pedicle screw loosening of non-augmented screws had a pooled risk of 22.5% (95% CI 10.8−36.6%, 95% prediction interval (PI) 0–81.2%) and augmented screws of 2.2% (95% CI 0.0–7.2%, 95% PI 0–25.1%) in patients with osteoporosis [20]. However, the incidence of pedicle screw loosening might be substantially higher, with up to 11% in patients with osteoporosis, according to the literature [21]. Calcium phosphate cements (CPCs) promise an advantageous alternative due to their unique properties, including bioactivity, osteoconductivity, and resorbability [22]. Hydroxyapatite (HAp, Ca_10_(PO_4_)(OH)_2_) and brushite (CaHPO_4_.2H_2_O) are formed by CPCs comprised of different combinations of calcium phosphate salts upon mixing with aqueous media [23]. The capability of α-tricalcium phosphate (α-TCP) to set into monolithic calcium-deficient hydroxyapatite at near physiological pH and temperature makes it a suitable candidate for bone cement formulation [23]. The combination of PMMA and CPCs has been shown to offer comparable stiffness in vertebral fracture restoration to commercial PMMA cement while compromising the osteoconductivity and ultimately sacrificing the resorbability of CPCs alone [24].

P-15, a synthetic amino-acid sequence identical to the alpha1 chain of type I collagen, has been demonstrated to possess osteoinductive properties when bound to a calcium matrix [25]. Synthetic P-15 has been shown to facilitate successful fusion in a prospective cohort study [26]. CPCs and P-15 offer similar advantages, such as osteoinductivity, osteoconductivity, and resorbability. Due to their similar characteristics, a combination of both materials promises synergistic effects while maintaining primary stability for implant augmentation. 

Our study aimed to formulate a novel bioactive, resorbable, and osteoinductive bone cement for pedicle screw augmentation in osteoporotic vertebrae using a combination of CPCs and P-15. Furthermore, to analyze its biomechanical properties, we developed a novel in vitro ovine osteoporosis model.

## 2. Materials and Methods

### 2.1. Cement Formulation

CPC/P-15 cement was formulated with a composition of 54% α-TCP, 16% dicalcium phosphate dehydrate (Sigma-Aldrich, St. Louis, MI, USA, and 30% P-15 (Cerapedics, Westminster, CO, USA) adsorbed to precipitated hydroxyapatite. The powder was liquified using 1% disodium hydrogen phosphate dodecahydrate. To each vertebra, 1.5 mL of the mixture was applied per screw.

### 2.2. Cell Culture

Human MES were derived as previously described [27]. Cells were cultured using α-MEM (Sigma Aldrich, St. Louis, Missouri, USA) together with 20% fetal bovine serum, 1 mM sodium pyruvate, penicillin, and streptomycin at 37 °C and 5% CO_2_. For co-culture assays, chamber slides (Thermo Fischer Scientific, Waltham, MA, USA) were coated with either CPC, P-15, or CPC/P-15. Up to 10^5^ cells were seeded per well and treated for ten days.

### 2.3. Immunostaining

Cells were rinsed with 4% neutral-buffered formalin (Sigma Aldrich, St. Louis, Missouri, USA) for fixation. Permeabilization was performed using 1% Triton X-100 (Sigma-Aldrich, St. Louis, Missouri, USA) in phosphate-buffered saline (Thermo Fisher Scientific, Waltham, MA, USA) for 10 min. Slides were then incubated with the primary antibody (1:100 in normal serum) overnight at 4 °C. For detecting the primary antibody, species-matched fluorophore-coupled antibodies were incubated for 1 h at room temperature. Slides were then covered with antifade mounting medium (Vectashield, Vector Laboratories, Burlingame, CA, USA) and coverslips were placed on top. All fluorescence and bright-field microscopy-based assays were observed using a Nikon Eclipse Ti microscope (Nikon, Tokyo, Japan).

### 2.4. Real-Time PCR

Total RNA for real-time PCR (RT-PCR) was extracted using TRIzol and treated with RNase-free DNase (QIAGEN, Hilden, Germany). The RNA concentration was quantified using a Nanodrop RNA 6000 (Thermo Fischer Scientific, Waltham, MA, USA) and analyzed using an Applied Biosystems StepOnePlus PCR machine (Thermo Fischer Scientific, Waltham, MA, USA). The mRNA expression analysis was carried out using Power SYBR Green (Applied Biosystems, Foster City, CA, USA). Base two was used to normalize the expressed values. 

### 2.5. Specimen Preparation and Screw Insertion 

For our study, a total of 40 ovine vertebrae from freshly frozen mature sheep spines were used. Immediately before use, vertebrae were defrosted and carefully dissected. The left pedicle was cannulated using a micro-speed drill. Commercially available, self-tapping titanium screws (Ulrich Medical, Ulm, Germany) were used for pullout tests. Only one screw was placed in each vertebra. The screws were manually inserted.

### 2.6. Decalcification

Vertebrae were dissected, and an intraosseous cannula (Teleflex, Wayne, PA, USA) was subsequently inserted in both pedicles. Vertebrae were flushed repeatedly using normal saline before decalcification. Decalcifying solution (Shandon TBD-1, Thermo Fischer Scientific, Waltham, MA, USA) in dilution 1:4 was perfused at 2 mL/h using a syringe pump (Braun, Kronberg, Germany) for 12 h via each pedicle. Vertebrae were rinsed and kept in normal saline. Measurements of the BMD were performed prior to and after decalcification. 

### 2.7. Imaging and Bone Mineral Density Measurement

Vertebrae were scanned using a computed tomography (CT) scanner (Aquilion Precision, Canon, Tokyo, Japan). Bone density (in Hounsfield Units, HU) was measured in the center of a mid-sagittal cross-section to provide a reference bone density of each vertebra tested. DEXA (Hologic Discovery, Hologic Inc, Waltham, MA, USA) was used to quantify the BMD of each vertebra (Apex 3.5.0.1).

### 2.8. Pullout Tests 

Vertebrae were fixed in an adjustable mounting block rigidly connected to the base plate of a universal testing machine equipped with a 3 kN load cell (Hegewald & Peschke, Nossen, Germany). Screws were axially pulled out from one bone of each pair. The midpoint of each screw head was aligned with the load axis of the testing machine to ensure pure axial loading in the pullout test. The pullout testing machine was operated in displacement control using a cross-head speed of 0.5 mm/s. The ultimate loads were determined as the maximum value from the load-displacement curves recorded. In the pullout tests, the maximum load was marked by a clear drop of the curve.

### 2.9. Statistical Analysis

All microscope-based assays were edited/quantified using ImageJ. Data are expressed as mean ± standard deviation (SD). Unpaired 2-tailed Student’s *t*-test was used for a comparison between two groups. Each group was tested for Gaussian distribution if one-way ANOVA was passed, followed by Bonferroni’s test. If this failed, a Kruskal–Wallis test followed by Dunn’s correction was conducted to test for significance among multiple groups. Statistical analyses were performed using Graph Pad Prism 6 software. *p* < 0.05 was considered statistically significant.

## 3. Results

### 3.1. Osteodifferentiation In Vitro

We first investigated the effect of each cement component on the osteodifferentiation of MESs. Cells were treated with either CPC, P-15, or a mixture of both. Untreated cells served as controls. Cells were cultured for 10 days. Characterization of osteodifferentiation was performed by RT-PCR, measuring the mRNA expression of ALPI II, osteopontin, RUNX2, collagen type-I, osteonectin, and osteocalcin. After treatment with CPC, the following were upregulated: ALPI II (22.91 ± 0.011-fold, *p* < 0.0001), osteopontin (0.51 ± 0.1-fold, *p* < 0.0001), RUNX2 (0.65 ± 0.06-fold, *p* < 0.0001), osteonectin (2.92 ± 0.1-fold, *p* < 0.0001), and osteocalcin (8.64 ± 0.41-fold, *p* < 0.0001). Treatment with P-15 lead to stronger osteodifferentiation, with upregulation of ALPI II (24.01 ± 2.48-fold, *p* < 0.0001), osteopontin (2.28 ± 0.7-fold, *p* < 0.0001), RUNX2 (2.36 ± 0.62-fold, *p* < 0.0001), collagen type-I (23.86 ± 4.11-fold, *p* < 0.0001), osteonectin (8.83 ± 2.1-fold, *p* < 0.0001), and osteocalcin (14.47 ± 3.87-fold, *p* < 0.0001). Treatment with CPC/P-15 lead to similar results, with the upregulation of ALPI II (33.27 ± 5.8-fold, *p* < 0.0001), osteopontin (6.15 ± 0.52-fold, *p* < 0.0001), RUNX2 (0.75 ± 0.1-fold, *p* < 0.0001), collagen type-I (8.5 ± 0.6-fold, *p* < 0.0001), osteonectin (12.12 ± 1.75-fold, *p* < 0.0001), and osteocalcin (16.43 ± 3.04-fold, *p* < 0.0001) (Figure 1).

To substantiate osteodifferentiation, immunofluorescence staining of osteopontin and staining of Ca2+ deposits using Alizarin Red were used. MESs were cultured in a growth medium and showed no osteopontin or Alizarin Red labeling. When treated with CPC/P-15, osteopontin and Ca2+ deposits, colored red by the Alizarin Red staining, were detected after 10 days (Figure 2A,B).

### 3.2. Axial Pullout Test and BMD in Non-Osteoporotic Bone

The BMD of the vertebrae of adolescent sheep was 0.72 ± 0.02 g/cm^2^. The BMD at the center of the mid-sagittal cross-section measured using CT was 511.8 ± 90.6 HU. Pullout strength was measured 12 h after pedicle screw insertion. F_max_ in untreated vertebrae was 1405 ± 56 N, in those with PMMA, 2112 ± 98 N (*p* < 0.0001), and 2010 ± 168 N (*p* < 0.0001) when augmented with CPC/P-15. There was no statistically significant difference between PMMA and CPC/P-15 treatments (*p* = 0.434; Figure 3).

### 3.3. Axial Pullout Test and BMD in Osteoporotic Bone

BMD in the vertebrae of adolescent sheep treated with a decalcifying solution was 0.53 ± 0.04 g/cm^2^. The BMD at the center of the mid-sagittal cross-section measured using CT was 101.3 ± 124.4 HU. The BMD was decreased by 26.17% ± 0.04% (Figure 4).

Pullout strength was measured 12 h after pedicle screw insertion. F_max_ in untreated vertebrae was 828 ± 66 N, in those with PMMA, 1324 ± 712 N (*p* = 0.04), and 1252 ± 131 N (*p* < 0.0078) in those augmented with CPC/P-15. There was no statistically significant difference between PMMA and CPC/P-15 (*p* = 0.94; Figure 5).

### 3.4. Discussion

As the prevalence of osteoporosis increases within the general population, so does its importance to spinal surgeons. Patients remain active for longer, so the proportion of spinal patients with osteoporosis continues to grow [28]. Previous studies indicated a relationship between low bone mass, subsidence of interbody devices, and higher non-union rates after spinal fusion [14,29]. This is plausible since vertebrae with a BMD below 80 mg/cm^3^ only reach 45% of normal bone strength, thus offering less pedicle screw stability [30]. The phenomenon is reflected in our in vitro model of osteoporotic ovine vertebrae. Here, we provide evidence that a decrease of the BMD of ~30% leads to a decreased axial pullout resistance of pedicle screws, to approximately 70% of their initial resistance. Different surgical techniques and medical anti-osteoporosis treatments have been propagated to improve pedicle screw stability. However, current evidence on the benefit of preoperative treatment remains ambivalent. Two studies using bisphosphonates as a preoperative treatment showed contradictory results with either higher or lower fusion rates [16]. Two studies published by Ohtori et al. found higher fusion rates and reduced pedicle screw loosening in postmenopausal women treated with Teriparatide before lumbar fusion surgery [31,32]. Cement augmentation of pedicle screws with PMMA proved to be beneficial regarding fusion rates after a three-year follow-up [17]. In a similar manner, augmentation with PMMA led to an increased pullout resistance in osteoporotic vertebrae in our in vitro model. Our data substantiate the clinical observation of increased implant failure after spinal fusion surgery in patients with osteoporosis, and provide evidence of the translational importance of our model for pre-clinical testing of spinal implants, bone substitutes, and cements. However, the use of PMMA comes with several serious downsides, such as cement embolism, extravasation with neuronal damage, and impairment of the bone micro-milieu due to exothermic polymerization. The latter diminishes the effects of osteoporotic medication and thus endangers long-term surgical outcomes. 

As PMMA further jeopardizes the potential benefit of osteoanabolic medication, we aimed to develop a biodegradable and osteoinductive bone cement, providing sufficient primary stability while enabling bone regeneration over time. The demanded optimal key features of such a bioactive cement in the clinical application are an adequate setting time, injectability, osteoinductive capacity, and biodegradation time [33]. In our study, we combined the osteoconductive and anabolic biomaterials CPC and synthetic collagen I mimetic P-15 to overcome the potentially harmful effects of PMMA. CPCs offer an alternative, since such types of cements set in the absence of an exothermic reaction, while also being osteoconductive and resorbable [18,33]. These characteristics provide a better aptitude for bone regeneration in the long term. However, some disadvantages of biodegradable cement compositions hindered the routine use in the clinical setting, such as the unfavorable imbalance of the concurrent biodegradation process in relation to bone regeneration and lower primary stability compared to PMMA, resulting in an inappropriate maintenance of vertebral body height [34,35]. These drawbacks need to be surmounted in cement-augmented procedures to achieve a stable primary situation and osteogenesis. Several recent experimental studies investigated alternative cement compositions to meet the aforementioned optimal requirements. The addition of magnesium and strontium led to improved self-setting properties and mechanical strength but lacked sufficient injectability [36]. Due to their low setting temperature and intrinsic porosity, CPCs are ideal drug delivery materials [37]. As CPCs are predominantly used in traumatology and dentistry, antibiotics are the most common drugs incorporated to date [38]. CPCs have often been used for vertebroplasty and kyphoplasty in the treatment of osteoporotic vertebral fractures. Here, the incorporation of bisphosphonates into CPCs improved the bone microarchitecture adjacent to the augmented bone defect in a large animal model of osteoporosis [39]. Another field of use is the direct coating of surgical implants, such as pedicle screws, using the main component of CPCs, HA. HA-coated pedicle screws showed a significantly higher extraction torque compared to uncoated screws, indicating an improved bone–implant interface and implant integration [40,41]. The presented data offer the first evidence of the feasibility of combining CPC and P-15 to achieve equal primary stability while maintaining osteoinductive properties in vitro.

Axial pullout tests of CPC/P-15-augmented pedicle screws implanted in sheep vertebra characterized by an average or low bone mineral density resulted in similar primary stability. These results fall in line with the previous reports of Moore et al. that outline how CPCs can provide similar primary stability for pedicle screw augmentation compared to PMMA [42]. Similar abilities of CPC were found regarding the pullout strength of osteoporotic canine vertebrae [43]. Likewise, CPC has been widely used for kyphoplasty and vertebroplasty to treat osteoporotic vertebral fractures. When compared with PMMA, no statistically significant difference in compression stiffness was detected [44]. Although CPC exhibits a comparable compression stiffness, its resistance against flexural, tractive, and shear forces is lower compared to PMMA [35]. These disadvantages curtail the routine use of CPC for either pedicle screw augmentation, kyphoplasty, or vertebroplasty. Due to their low-temperature setting reaction and intrinsic porosity, CPCs might be used as drug-delivery materials [37]. Efforts have been made to use bisphosphonate-loaded CPC for vertebroplasty in an osteoporotic sheep model. Analysis of microarchitectural parameters in vertebrae augmented with such loaded CPC demonstrated a positive influence on the microarchitecture of the adjacent trabecular bone [39]. To our knowledge, the loading of CPCs with synthetic P-15 has not been described before. We provide evidence that this combination yields the potential for bone formation to increase BMD, and thus mitigates the current shortcomings of CPCs in vertebral and pedicle screw augmentation.

The remodeling cycle of bone begins early in life and depends on the coordinated interaction of two cell lineages facilitating bone resorption (osteoclasts) and the deposition of new bone material (osteoblasts) [45]. Both osteoclasts and osteoblasts derive from mesenchymal stem cells (MESs) during osteogenic differentiation [46]. Osteoblast differentiation is a three-step process, taking several weeks in vitro [47]. Downregulation of DNA replication coincides with the expression of osteoblast markers such as alkaline phosphatase (ALPI II), collagen I, and RUNX2 [48]. First, proliferation and the expression of fibronectin, collagen, transforming growth factor β receptor 1 (TGFβ1), and osteopontin (step 1) occurs, before the cell exits the cell cycle and differentiation begins, while maturating the extracellular matrix with ALPI II and collagen (step 2) and matrix mineralization with osteocalcin (step 3) [48]. We investigated this entire developmental course in different treatment groups. RT-PCR showed a significant upregulation of osteoblast-associated genes in MES. Furthermore, bone formation was found as indicated by Ca^2+^ deposit formation in vitro. While the osteoinductive features of P-15 have been described in some previous studies, no data are available on the effects of P-15 in combination with CPC [49]. Thus, our data present the first evidence of the feasibility of using P-15 as an additive in bioactive bone cements. Our findings suggest that the combination of both substances might have synergistic effects in terms of ALPI II and osteocalcein expression. These results prove the capability of the CPC/P-15 composition to promote osteoblastic as well as the subsequent osteoclastic differentiation. Nevertheless, gene expression analysis in vitro provides only a snapshot of ongoing physiological processes and might not translate in vivo. The observed additive effects might be partially caused by differences in cell culture conditions and need to be verified in an animal model.

## 4. Conclusions

P-15-loaded CPC can induce osteodifferentiation in human MESs in vitro, while providing similar pullout strength compared to PMMA. In our ovine osteoporosis model, BMD and the decrease in pullout loads of pedicle screws were identical to what is expected from osteoporotic vertebrae in vivo. In addition, CPC-P-15 provided similar primary stability for pedicle screw augmentation compared to PMMA in our osteoporosis model. However, higher costs and inferior injectability limit its clinical application. Further research is necessary to exploit the full biological potential of CPC-P-15 on a daily clinical basis. 

## Figures and Tables

**Figure 1 biomedicines-09-01392-f001:**
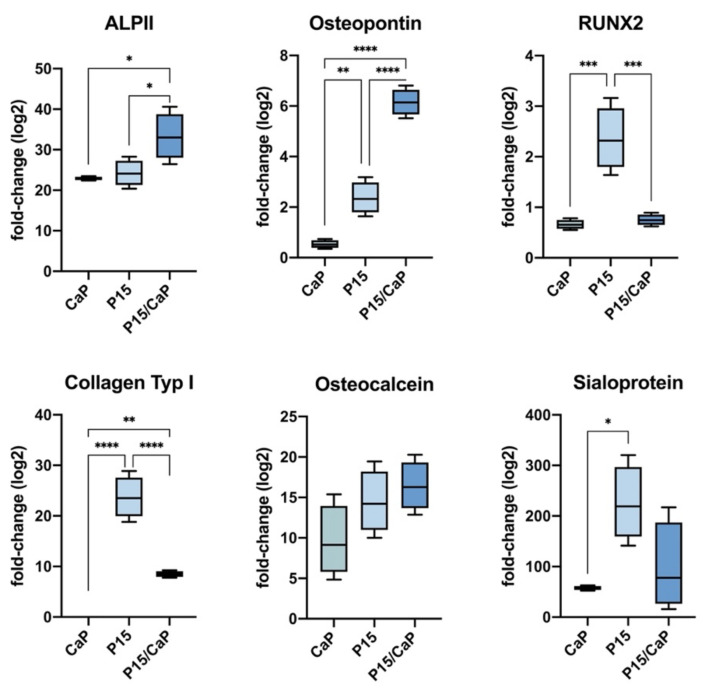
Osteodifferentiation of MESs in vitro. Using RT-PCR, mRNA expression of ALPI II (22.91 ± 0.011-fold, *p* < 0.0001), osteopontin (0.51 ± 0.1-fold, *p* < 0.0001), RUNX2 (0.65 ± 0.06-fold, *p* < 0.0001), osteonectin (2.92 ± 0.1-fold, *p* < 0.0001), and osteocalcin (8.64 ± 0.41-fold, *p* < 0.0001) were upregulated after treatment with CPC-P-15. * *p* < 0.05, ** *p* < 0.01, *** *p* < 0.005, **** *p* < 0.0005.

**Figure 2 biomedicines-09-01392-f002:**
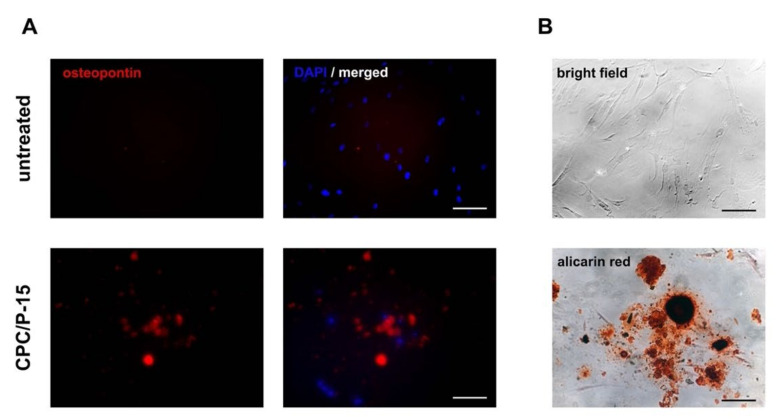
(**A**) Immunofluorescence staining of osteopontin and staining of Ca2+ deposits. (**B**) MESs cultured in a growth medium showed no osteopontin or Alizarin Red labeling When treated with CPC/P-15, osteopontin and Ca2+ deposits, colored red by the Alizarin Red staining, were detected after 10 days.

**Figure 3 biomedicines-09-01392-f003:**
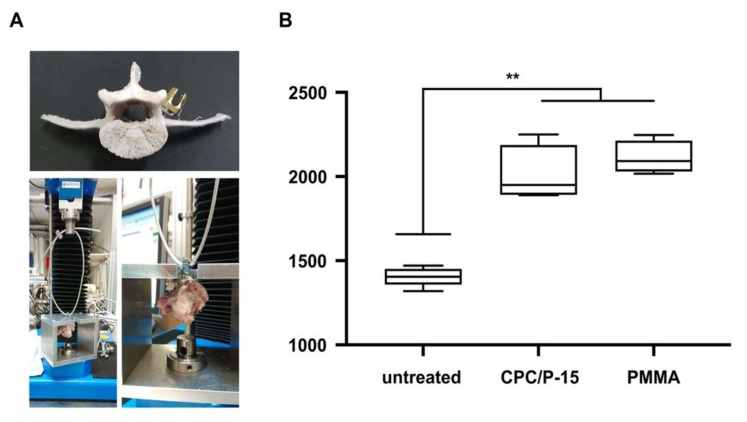
Axial pullout test and BMD in non-osteoporotic bone. (**A**) Pedicle screw placement in a skeletonized vertebra and set-up for pull-out testing. (**B**) F_max_ in untreated vertebrae was 1405 ± 56 N, in those with PMMA, 2112 ± 98 N (*p* < 0.0001), and 2010 ± 168 N (*p* < 0.0001) when augmented with CPC/P-15. There was no statistically significant difference between PMMA and CPC/P-15 treatments (*p* = 0.434). ** *p* < 0.01.

**Figure 4 biomedicines-09-01392-f004:**
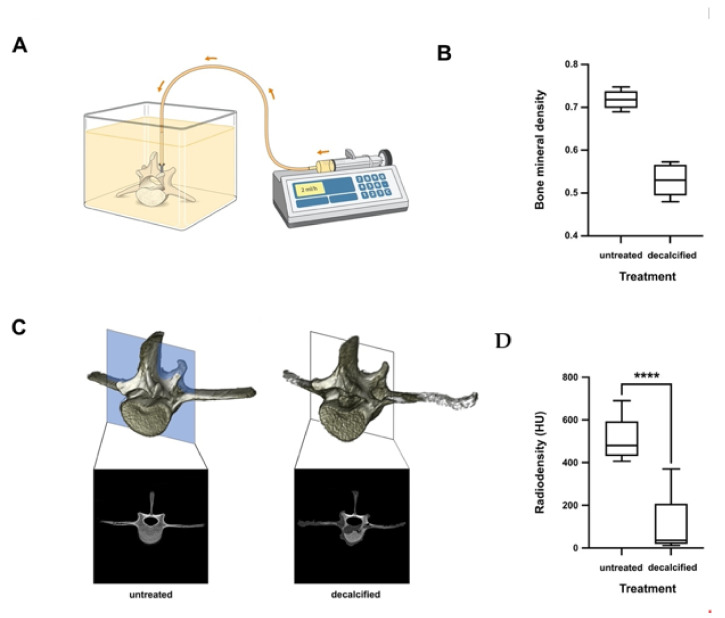
(**A**) Ovine osteoporosis model in vitro. (**B**) The BMD of the vertebrae of adolescent sheep was 0.72 ± 0.02 g/cm^2^. (**C**,**D**) The BMD at the center of the mid-sagittal cross-section measured using CT was 511.8 ± 90.6 HU. BMD in the vertebrae of adolescent sheep treated with a decalcifying solution was 0.53 ± 0.04 g/cm^2^. The BMD at the center of the mid-sagittal cross-section measured using CT was 101.3 ± 124.4 HU. The BMD was decreased by 28.75% ± 2.6%. **** *p* < 0.0005.

**Figure 5 biomedicines-09-01392-f005:**
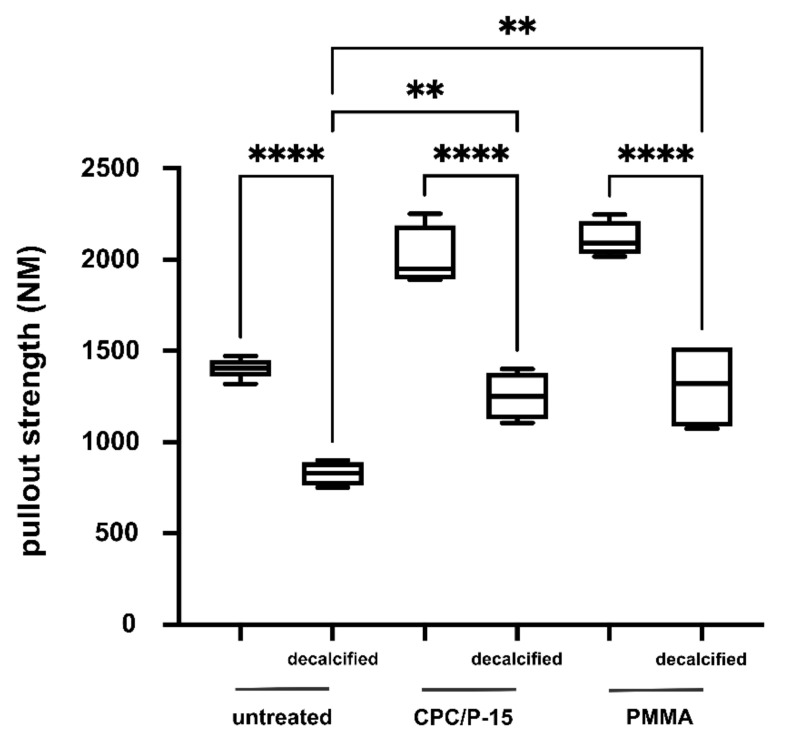
Axial pullout test and BMD in osteoporotic bone. In decalcified vertebrae, F_max_ was significantly lower (1405 ± 55.97 N) compared to not decalcified vertebrae (828 ± 66 N; *p* < 0.0001). F_max_ was significantly higher in those decalcified vertebrae augmented with CPC/P-15 (1252 ± 131 N; *p* < 0.0078) and those with PMMA (1308 ± 244 N; *p* = 0.04). There was no statistically significant difference between PMMA and CPC/P-15 (*p* = 0.94). ** *p* < 0.01, **** *p* < 0.0005.

## Data Availability

Not applicable.

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
