# Peer review of "Novel Biodegradable Composite of Calcium Phosphate Cement and the Collagen I Mimetic P-15 for Pedicle Screw Augmentation in Osteoporotic Bone"

_biomedicines, 2021, doi:10.3390/biomedicines9101392_

Round 1

Reviewer 1 Report

The reviewer’s comments are as below:

  • The manuscript is written very well with proper scientific language. However, the research study is lacking novelty. Various researchers have already worked on introducing Calcium phosphate cement (CPCs) and P-15 individually for specific purposes. This research has just combined the two techniques to investigate the synergistic effect. This further raises the question to mix CPCs and P-15 with PMMA to formulate biocomposite materials for the same application. Please clarify it in the introduction section.
  • As per Figure 4, The BMD was decreased by 28.75%. Please include an explanation of how it was calculated in the results section. Not sure, but the value appears to be 26.38% and needs a reconfirmation of the correct calculation.
  • As per Figure 5, Fmax in un-treated vertebrae is 828±66 N, in PMMA it is 1324±712 N, and in CPC/P-15 it is 1252±131 N. If PMMA and CPC/P-15 are not significantly different, then considering a high value of standard deviation in PMMA, this can also be statistically not significant for the untreated sample as well. Please clarify and provide a p-value for the untreated sample.
  • In the section “Discussion”, line 4: Please clarify “non-un-ion” rates, is it Non-Union rates?
  • Considering the drawback of this newly formulated material compared to the PMMA based material, it would be good to include a future research study recommendation in the “Conclusion” section based on the current findings.

Author Response

We would like to thank the reviewer for his insightful remarks. His comments have been addressed carefully and the manuscript updated accordingly. We believe that these changes strengthen the manuscript substantially.

The manuscript is written very well with proper scientific language. However, the research study is lacking novelty. Various researchers have already worked on introducing Calcium phosphate cement (CPCs) and P-15 individually for specific purposes. This research has just combined the two techniques to investigate the synergistic effect. This further raises the question to mix CPCs and P-15 with PMMA to formulate biocomposite materials for the same application. Please clarify it in the introduction section.

-> Further explanations have been added to the introduction:

1) The combination of PMMA and CPCs has been shown to offer comparable stiffness in vertebral fracture restoration to commercial PMMA cement while compromising the osteoconductivity and ultimately sacrificing the resorbability of pure CPCs.

2) CPCs and P-15 offer similar advantages such as osteoinductivity, osteoconductivity and resorbability. Due to their similar characteristics a combination of both materials promises synergistic effects while maintaining primary stability for implant augmentation.

3) Our study aimed to formulate a novel bioactive, resorbable and osteoinductive bone cement for pedicle screw augmentation in osteoporotic vertebrae using a combination of CPCs and P-15.

As per Figure 4, The BMD was decreased by 28.75%. Please include an explanation of how it was calculated in the results section. Not sure, but the value appears to be 26.38% and needs a reconfirmation of the correct calculation.

-> The calculation has been corrected to 26.17±0.04%.

As per Figure 5, Fmax in un-treated vertebrae is 828±66 N, in PMMA it is 1324±712 N, and in CPC/P-15 it is 1252±131 N. If PMMA and CPC/P-15 are not significantly different, then considering a high value of standard deviation in PMMA, this can also be statistically not significant for the untreated sample as well. Please clarify and provide a p-value for the untreated sample.

-> There has been a mix up with an older figure. Updated figure 5 now depicted pull out strength (Fmax) for not augmented vertebrae either not decalcified or decalcified and the corresponding values for those augmented with CPC/P15 and PMMA. P values are given for each comparison.

In the section “Discussion”, line 4: Please clarify “non-un-ion” rates, is it Non-Union rates?

-> This seems to be a typo related to the manuscript submission, as it is not to been found in our manuscript.

Considering the drawback of this newly formulated material compared to the PMMA based material, it would be good to include a future research study recommendation in the “Conclusion” section based on the current findings.

-> The following sentences have been added to conclusions: “However, higher costs and inferior injectability limits its clinical application.  Further research is necessary to exploit the full biological potential of CPC-P-15 on a daily clinical basis.”

Reviewer 2 Report

This paper reports biological and biomechanical properties of a biodegradable composite used to improve pedicle screw stability in anti-osteoporotic treatment. The new bioresorbable composite, based on phosphate cements and a synthetic peptide, is an alternative to PMMA, not resorbable material leading to in situ chronic inflammation as a function of time.

To my opinion, the subject is appropriate for publication in Biomedecines: if the biological and biomechanical tests are well described, however the Material and Methods section is incomplete. In fact the formulation of composite was not reported (patented material, may be?), and there is no information on commercial origins of HAp and a-TCP (important in order to evaluate biomechanical results) or the percentage of organic part (collagen I mimetic P15 peptide) vs mineral components as well as the amount used in biological experiments.

Author Response

We would like to thank the reviewer for his insightful remarks. His comments have been addressed carefully and the manuscript updated accordingly. We believe that these changes strengthen the manuscript substantially.

This paper reports biological and biomechanical properties of a biodegradable composite used to improve pedicle screw stability in anti-osteoporotic treatment. The new bioresorbable composite, based on phosphate cements and a synthetic peptide, is an alternative to PMMA, not resorbable material leading to in situ chronic inflammation as a function of time.

To my opinion, the subject is appropriate for publication in Biomedecines: if the biological and biomechanical tests are well described, however the Material and Methods section is incomplete. In fact the formulation of composite was not reported (patented material, may be?), and there is no information on commercial origins of HAp and a-TCP (important in order to evaluate biomechanical results) or the percentage of organic part (collagen I mimetic P15 peptide) vs mineral components as well as the amount used in biological experiments.

-> CPC/P-15 cement was formulated as composition of 54% α-TCP, 16% dicalcium phosphate dehydrate and 30% P-15 adsorbed to precipitated hydroxyapatite. The powder was liquified using 1% disodium hydrogen phosphate dodecahydrate. 1,5ml of the mixture was applied to each vertebra per screw.

Reviewer 3 Report

A very interesting paper suitable for publication. To help the reader they should address the following two matters

a) p3 "The screws were manually inserted". I wondered why the authors did not measure the torque required to insert the screws. There are several reports on the correlation between insertion torque and pull out strength and provide a route to understanding why the treated screws exhibited a greater spread in results than the untreated.

b)p3 "using a cross-head speed of 0.5mm/s". The authors should comment on the choice of pullout speed and how they arrived at this selection

Author Response

We would like to thank the reviewer for his insightful remarks. His comments have been addressed carefully and the manuscript updated accordingly. We believe that these changes strengthen the manuscript substantially.

A very interesting paper suitable for publication. To help the reader they should address the following two matters

  1. a) p3 "The screws were manually inserted". I wondered why the authors did not measure the torque required to insert the screws. There are several reports on the correlation between insertion torque and pull out strength and provide a route to understanding why the treated screws exhibited a greater spread in results than the untreated.

-> We thank the reviewer for this insightful remark. Egol et al (Trauma 2004) and Ricci et al (JOT 2010) estimated the amount of insertion torque required for implant stability to be at least 3 Nm. It is true that the screws in our experiments were placed manually, and insertion torque was not measure. However, all screws were placed by the same researcher enabling for similar insertion torques and trajectories. Further, the scope of our study was not to determine absolute pull-out strength but rather a comparison between the pull-out strength of similar screws with different augmentation modalities. We believe that measurement of the insertion torque is therefor of lesser importance albeit still of high interest.

b)p3 "using a cross-head speed of 0.5mm/s". The authors should comment on the choice of pullout speed and how they arrived at this selection

-> The used cross-head speed of 0.5mm/min. The given time frame of sec presents a typo. The cross-head speed of 0.5mm/min wis based on previous research conducted be Sun et al (International Journal of Nanomedicine 2017). Here, the capability of injectable CPC for augmentation of pedicle screws has been analyzed. Cross head speeds of 0.5 to 1mm/min are commonly used throughout the literature for linear pedicle screw pull-out tests.